# Melatonin Attenuates RANKL-Induced Osteoclastogenesis via Inhibition of Atp6v0d2 and DC-STAMP through MAPK and NFATc1 Signaling Pathways

**DOI:** 10.3390/molecules27020501

**Published:** 2022-01-14

**Authors:** Seong-Sik Kim, Soon-Pill Jeong, Bong-Soo Park, In-Ryoung Kim

**Affiliations:** 1Department of Orthodontics, Dental Research Institute, Pusan National University Dental Hospital, Yangsan 50612, Korea; softid@pusan.ac.kr (S.-S.K.); spjeong@gmail.com (S.-P.J.); 2Department of Oral Anatomy, School of Dentistry, Pusan National University, Yangsan 50612, Korea; parkbs@pusan.ac.kr; 3Dental and Life Science Institute, School of Dentistry, Pusan National University, Yangsan 50612, Korea

**Keywords:** melatonin, bone remodeling, osteoporosis, V-ATPase, osteoclastogenesis

## Abstract

Melatonin is a hormone secreted by the pineal gland that is involved in the biorhythm of reproductive activities. The present study investigated the inhibitory effects of melatonin on osteoclastogenesis in RAW 264.7 cells according to changes in V-ATPase and the corresponding inhibition of the MAPK and NFATc1 signaling processes. Methods: the cytotoxic effect of melatonin was investigated by MTT assay. Osteoclast differentiation and gene expression of osteoclast-related factors were confirmed via TRAP staining, pit formation assay, immunofluorescence imaging, western blot, and real-time PCR. Results: melatonin was found to inactivate the p38 and JNK of MAP kinase in RAW264.7 cells treated with RANKL and treated with a combination RANKL and melatonin for 1, 3, and 5 days. The melatonin treatment group showed a reduction in osteoclastogenesis transcription factors and ATP6v0d2 gene expression. Conclusions: melatonin inhibits osteoclast differentiation and cell fusion by inhibiting the expression of Atp6v0d2 through the inactivation of MAPK and NFATc1 signaling in RANKL-stimulated RAW264.7 macrophages. The findings of the present study suggest that melatonin could be a suitable therapy for bone loss and imply a potential role of melatonin in bone health.

## 1. Introduction

Melatonin is a small indole molecule synthesized from serotonin in the pineal gland [1,2]. It is involved in a variety of physiological processes, including biorhythms, reproduction, aging, and the cardiovascular and immune systems. Many studies have reported on the pro-inflammatory, neuroprotective, and immunomodulatory activities of melatonin and its role in the bone healing process [3]. Melatonin controls the overall remodeling process through the dual action of osteoblast bone formation and osteoclast bone resorption [4,5]. In addition, in vitro and in vivo studies provide evidence that melatonin treatment contributes to the prevention of bone loss [6]. Specifically, it regulates the balance of osteoclasts and osteoblasts through signaling transduction, including the Wnt, NFκB, and MAPK pathways, to improve bone quality [7,8,9]. Older people tend to have decreased melatonin secretion, which is of interest to researchers studying osteoporosis because it is associated with an increased risk of osteoporosis [1,6]. Accordingly, melatonin intake may be an alternative therapy for improving bone health [10].

Bones have dynamic functions in the body, with significant metabolic, structural, and mineral-supporting roles. Over the human lifespan, bones are formed by regenerated tissues that have undergone remodeling through the action of bone resorption by osteoclasts and bone formation by osteoblasts [11,12]. Through the latter process, 10% of total bone is replaced by new bone and continuously updated during the human lifespan [13,14]. Bone remodeling is regulated by cytokines, such as TGF-β, TNF-α, and IL-10, and by calcium-regulating hormones, such as parathormone (PTH), thyroxine, and estrogen, among others [15,16,17]. These factors are important in osteoporosis and in skeletal changes associated with immobilization, inflammation, and Paget’s disease. 

Bone resorption is involved in a variety of physiological functions, including the removal of calcified cartilage during bone growth, modeling of bone during growth or adaptation, maintenance of mineral deterioration, removal of damaged skeleton, tooth rash, and orthodontic tooth movement [18,19]. Osteoclasts are multinucleated giant cells responsible for bone resorption and mature cells are involved in the activation of bone resorption. Osteoclast differentiation requires the cell-to-cell interaction of osteoclast precursor cells and osteoblasts [20], which is controlled by the presence of receptor activators for the nuclear factor-κB ligand (RANKL) [21]. RANKL plays a major role in the activation of osteoclastogenesis transcription factors [21,22]. The nuclear factor of activated T cells (NFATc1) is a major transcription factor that regulates a series of osteoclastogenesis-specific genes, including tartrate-resistant acid phosphatase (TRAP), cathepsin K, and calcitonin receptor [23,24]. Osteoclasts have specialized cell membrane areas called the ruffled border that bind to actual bone resorption sites on the bone surface. The resorption lacunae underneath the ruffled border membrane are acidic because the osteoclast ruffled border contains a vacuolar proton pump and adenosine triphosphatases (V-ATPases), which are vital for osteoclast acidification and function [25]. The acidic pH environment then dissolves bone mineral. In addition, proteinases capable of collagen degradation are active and present in acidic pH [26]. Several recent studies have reported the effect of melatonin on the differentiation of osteoclasts. These studies revealed that melatonin inhibits osteoclast differentiation through downregulation of the NF-κB pathway, along with reduction of RANKL-induced TRAF6, JNK, PRMT1 and NFATc1 transcription factor induction [6,27,28].

However, the effect of melatonin on osteoclast formation and cell fusion by V-ATPase through MAPK and NFATc1 signaling in the RAW264.7 mouse macrophage cell line is not currently known. In the present study, we present for the first time the finding that melatonin attenuates osteoclast differentiation and fusion through the inhibition of p38/JNK, NFATc1, and Atp6v0d2.

## 2. Results

### 2.1. Melatonin Inhibits RANKL-Induced Osteoclast Formation and Resorption Pit Formation in RAW 264.7 Cells

Before osteoclast differentiation, MTT assays were performed to observe whether the cell viability of melatonin affected RAW 264.7 macrophages. Each cell line was treated with 0–2000 μM of melatonin for 0.5 to 5 days. Melatonin gradually reduced the cell viability of the RAW 264.7 cell line from 67.05% (1000 μM, 5 days) to 45.36% (2000 μM, 5 days) (Figure 1A). To find the osteoclast formation and resorption, we used an osteoclast marker as tartrate-resistant acid phosphate (TRAP) staining and formation of resorption pits on the calcium coating Osteo assay plate. Melatonin (0, 100, 200, 300, 400, and 500 μM), which did not exhibit cytotoxicity, was then applied to RAW264.7 cells to observe the inhibitory effect on RANKL-induced osteoclastogenesis. We found that melatonin clearly reduced the number and size of osteoclast formations (TRAP-positive cells) (Figure 1C,E), osteoclast nuclei (Figure 1F), and resorption pits (Figure 1D,G) in a dose-dependent manner. In osteoclasts, actin rings play an important role in the attachment of osteoclasts to the bone surface and are considered hallmarks of mature osteoclasts [29]. We performed immunofluorescent staining to assess actin ring formation in mature osteoclasts. The RANKL alone treatment group showed a strong red fluorescence signal at the edge of the wrinkled membrane. However, the formation of actin rings was not observed in the combination treatment of RANKL and melatonin (Figure 1B).

### 2.2. Melatonin Inhibits MAPK and NFATc1 Signaling Pathway on RANKL-Induced Osteoclastogenesis

To compare the activity of MAPKs, such as extracellular signal-regulated kinase (ERK), p38, and c-Jun N-terminal kinase (JNK), the protein levels were determined by western blotting. RAW 264.7 cells were treated with RANKL (10 ng/mL) and RANKL + melatonin (300 μM) for a minimum of 5 min to a maximum of 60 min. The activity of phosphorylated ERK was significantly increased in the 5 min after RANKL + melatonin treatment (Figure 2A). However, the phosphorylation of p38 was significantly reduced in the RANKL + melatonin (300 μM) treatment group compared to the RANKL treatment groups (5, 10, and 15 min treatment groups; Figure 2B), and the phosphorylation of JNK was also significantly reduced significantly in the RANKL + melatonin (300 μM) treatment group (15 min, Figure 2C). These results suggest that melatonin can inactivate the p38 and JNK of MAP kinase, which are associated with the early RANKL-induced signaling pathways of RAW 264.7 cells, and can also inhibit the RANKL-mediated differentiation of these cells.

To determine the effect of melatonin on osteoclastogenesis, we investigated the protein expression of TRAF6, NFATc1, DC-STAMP, c-Fos, MMP-9, and cathepsin K (CTSK) using western blotting, immunofluorescence microscopy, and real-time PCR. RAW264.7 cells were treated with RANKL and RANKL + melatonin for 1, 3, and 5 days. As shown in Figure 3, the RANKL + melatonin treatment reduced TRAF6 (Figure 3A) and NFATc1 (Figure 3C) compared to the RANKL treatment group. The immunofluorescence results confirmed that TRAF6 (Figure 3B) and NFATc1 (Figure 3D) had a lower expression in the cells treated with RANKL + melatonin. The protein expression of DC-STAMP, MMP-9, c-Fos, and CTSK of cells treated with RANKL + melatonin was lower than those treated with RANKL alone (Figure 4). Therefore, these results indicate that melatonin exerts suppressive effects on osteoclastogenesis.

### 2.3. Melatonin Inhibits V-ATPase Vo Domain (Atp6v0d2) and DC-STAMP Expression

The activation of MAPK and NFATc1 signaling transduction is involved in osteoclast formation by RANKL [30]. We previously confirmed that melatonin regulates the expression of osteoclast marker genes through the suppression of p38, JNK, and NFATc1 signaling (Figure 2 and Figure 3). ATP6v0d2 is known to be an essential component of osteoclast-specific proton pumps that mediate extracellular acidification in bone resorption [31]. We next confirmed the expression of ATP6v0d2 and activity of V-ATPase during osteoclastogenesis in RAW264.7 cells treated with RANKL (10 ng/mL) and melatonin (100 to 500 μM) for 1 to 5 days. The latter treatment reduced ATPase activity (day 0, 897 pmol; day 1, 875 pmol; day 3, 1101 pmol; day 5, 775 pmol) (Figure 5B) and ATP6v0d2 gene expression (Figure 5C). To compare the effect of melatonin on the ATP6v0d2 protein and gene expression, RAW264.7 cells were treated with RANKL and RANKL + melatonin and incubated for 1, 3, and 5 days (Figure 5A,D). RANKL alone markedly increased both the ATP6v0d2 protein and gene expression, whereas RANKL + melatonin inhibited Atp6v0d2 expression during osteoclastogenesis. To investigate whether melatonin inhibits ATP6V0d2 and DC-STAMP through the signaling pathway of NFATc1, we performed siRNA-mediated stable knockdown of NFATc1 expression in cultured RAW264.7 cells. Expression was significantly inhibited at the protein level by si-NFATc1 (Figure 5E, western blotting, second line). Moreover, the expression levels of NFATc1, DC-STAMP, and ATP6V0d2 were more significantly reduced in the melatonin-treated group (Figure 5E, western blotting, fourth line, transfection of si-NFATc1 + melatonin 300). These data clearly suggest that melatonin inhibits the expression of Atp6v0d2 and DC-STAMP through the inactivation of MAPK and NFATc1 signaling, thus reducing osteoclast differentiation and maturation.

## 3. Discussion

Many kinds of stimuli, including hormones, cytokines, and mechanical stimuli, affect bone turnover, which in turn affects the amount and quality of tissue produced [32]. Different studies on hormone replacement therapy have been conducted to treat bone diseases. The parathyroid hormone, for example, has been shown to increase bone density in diseases such as osteoporosis [33]. It is also known that melatonin secreted from the pineal gland affects osteogenesis and numerous studies on this phenomenon have been carried out [10,34,35] or are currently underway. When osteoclasts and osteoblasts are dynamically balanced, bone metabolism progresses. When osteoclasts are excessive, bone resorption occurs and bone formations weaken, leading to osteolysis [36]. The ideal therapy for bone disease is to promote osteoblast differentiation and to inhibit osteoclast formation and destruction, thereby improving bone quality [37]. Therefore, it is important to find drugs that have a positive effect on bone metabolic diseases and to elucidate the mechanisms of such drugs on osteoclasts or osteoblasts. In the present study, we demonstrate for the first time that melatonin inhibits Atp6v0d2 through the MAPK and NFATc1 signaling pathways. Recently, melatonin was used in clinical trials in postmenopausal women with exogenous administration of melatonin (Melatonin Osteoporosis Prevention Study (MOPS); NCT01152580); in postmenopausal women with osteopenia (Treatment of Osteopenia With Melatonin (MelaOST), NCT01690000 and Melatonin-Micronutrients for Osteopenia Treatment Study (MOTS), NCT01870115), effects such as improvement of bone mineral density (BMD) (MelaOST, MOTS) and bone marker conversion status (MOPS, MOTS) were confirmed throughout clinical and preclinical studies [38]. Therefore, it is considered necessary to identify the signaling pathway of melatonin in women entering menopause in the future and to pursue research on changes in ATP6v0d2 and DC-STAMP activity.

Osteoclasts produce an acidic microenvironment where protease release activity causes bone demineralization and substrate degradation [39]. TRAP is used as a histochemical marker of osteoclasts, as it is expressed by osteoclasts, macrophages, dendritic cells, and many other cell types. In addition, TRAP plays a crucial role in many biological processes, including skeletal formation, collagen synthesis and degradation, bone mineralization, and cytokine production by macrophages [40]. To form a complex with RANKL and its receptor RANK, it is essential for bone marrow progenitor cells or macrophages to differentiate into mature osteoclasts [41]. RANKL uses MAPKs (ERK, JNK, p38) as a signaling mediator as a major regulator of osteoclast formation, and immediate (5–20 min) phosphorylation [42]. These essential interactions lead to the activation of MAPKs and NF-κB, which induce the self-amplification of NFATc1; then, NFATc1 translocates to the nucleus and promotes the expression of osteoclast-associated TRAP, MMP9, CTSK, DC-STAMP, and OSCAR genes [43]. In the present study, melatonin significantly reduced the number and shape of TRAP-positive osteoclasts in the early stages of osteoclast differentiation and weakened the resorption of calcium phosphate. Overall, our results show that melatonin can affect osteoclast differentiation as well as osteoclast bone resorption. Melatonin also inhibited the expression of osteoclast-associated factors such as NFATc1, TRAF6, cFos, MMP9, CTSK, and DC-STAMP, and phosphorylated forms of MAPKs such as p38 and JNK, except for Erk. It is inferred that p38 and JNK among MAPKs were targets of melatonin, which suppressed their expression, followed by suppression of other osteoclastogenic factors. However, the study of Ping et al. [8] found that the osteoclast differentiation inhibitory effect of melatonin occurred through the abolition of the NF-kB signaling pathway, but that the MAPK and PI3 K/AKT signaling pathways did not show significant changes. 

Very recently, it was also reported that melatonin inhibited tumor-associated osteoclast formation via the ERK and p65 pathways, whereas it did not affect p38 or JNK activity [28]. These different findings are still controversial because they conflict with this study. Mononuclear osteoclast precursors fuse to become mature functional osteoclasts. In particular, v-ATPase Vo domain d2 isoform (Atp6v0d2) and the dendritic cell specific transmembrane protein (DC-STAMP) are important regulators of osteoclast fusion and maturation [44,45]. In particular, Atp6v0d2 is a factor highly expressed in osteoclasts and Atp6v0d2 is known to be related to osteoclast maturation and cell-cell fusion, not differentiation [46]. Recently, Bae et al. showed that melatonin-treated PDLC and cementoblasts indirectly regulate osteoclast differentiation. Specifically, melatonin significantly reduced the formation and number of osteoclasts and the expression of osteoclast markers such as DC-STAMP, c-fos, and NFATc1 [47]. However, how melatonin directly affects the expression of Atp6v0d2 and DC-STAMP during osteoclast fusion and maturation has yet to be elucidated. To determine whether melatonin directly affects the expression of Atp6v0d2 and DC-STAMP and osteoclastic cell-cell fusion, we evaluated ATPase activity as well as the expression of ATP6v0d2 and DC-STAMP. Melatonin markedly reduced ATPase activity and the protein and gene expression of ATP6v0d2 and DC-STAMP during the osteoclast process induced by RANKL. In this study, it was confirmed that the expression of NFATc1 and ATP6v0d2 was suppressed by melatonin treatment. In addition, we found that NFATc1 knockdown using si-RNA decreased the total amount of DC-STAMP and ATP6V0d2, and showed a more significant decrease in the melatonin-treated group (Figure 5E). This suggests that melatonin targets NFATc1 and leads to a decrease in ATP6v0d2 and DC-STAMP due to inhibited NFATc1.

In the present study, we demonstrated for the first time that melatonin inhibits osteoclast differentiation, cell-cell fusion, and maturation by inhibiting the expression of Atp6v0d2 and DC-STAMP through the inactivation of MAPK and NFATc1 signaling in RANKL-stimulated RAW264.7 macrophages. Therefore, the findings of the present study confirm that melatonin may be a suitable therapy for bone loss and delineate the potential role of melatonin in bone health.

## 4. Materials and Methods

### 4.1. Cell Culture

The murine macrophages, RAW 264.7, were purchased from the Korea Cell Line Bank (KCLB; Seoul, Korea). RAW264.7 cells were subcultured every 2 days using a medium (Dulbecco Modified Eagle’s Medium, DMEM; Hyclone, Logan, UT, USA) containing 10% fetal bovine serum (FBS; Hyclone, Logan, UT, USA) in 5% CO_2,_ 37 °C incubators. For osteoclast differentiation, the culture environment of RAW264.7 cells were seeded not to exceed 20% of the total area of the culture plate, and a differentiation medium containing 10 ng/mL of RANKL (DMEM including 10% FBS) was applied and cultured for 5 days. The differentiation medium was changed daily for 5 days.

### 4.2. Cell Viability Assay

RAW 264.7 cells (1 × 10^4^ cells/well) were seeded in a 96-well plate and incubated for 2 h in a CO_2_ incubator to allow the cells to adhere to the bottom of the well. After that, various concentrations of melatonin (10~2000 μM) were treated and incubated for 12 h to 24 h. The supernatant was removed from each well, and 100 μL of the medium containing MTT (1-(4,5-Dimethylthiazol-2-yl)-3,5-diphenylformazan, Sigma-Aldrich, Louis, MO, USA) (0.5 mg/mL) was added to each well and reacted in a 37 °C incubator for 4 h. The MTT solution was removed from each well, and 100 microliters of DMSO was added to dissolve the purple formazan crystals and reacted for 10 min on a shaker. Finally, the absorbance of each well was measured at a wavelength of 570 nm using a SpectraMax iD3 microplate reader (Molecular Devices, Sunnyvale, CA, USA). Data were derived from at least three independent experiments.

### 4.3. Tartrate-Resistant Acid Phosphatase (TRAP) Staining

RAW 264.7 cells (5 × 10^4^ cells/well) were seeded in a 24-well tissue culture plate and incubated for 2 h in a CO_2_ incubator to allow the cells to adhere to the bottom of the well. Then, cells were applied to a differentiation medium containing 10 ng/mL of RANKL (DMEM including 10% FBS) and treated with various concentrations (100 to 500 μM) of melatonin. The medium was replaced daily for 5 days. After incubation, cells were fixed in 4% formaldehyde solution (Sigma-Aldrich, Louis, MO, USA) for 10 min and then stained using the Acid Phosphatase, Leukocyte (TRAP) Kit (Sigma-Aldrich, Louis, MO, USA) according to the manufacturer’s instructions. TRAP-positive cells were stained purple and confirmed with an Olympus CKX41 inverted optical microscope (Olympus, Tokyo, Japan) equipped with a digital camera (Nikon Coolpix, Tokyo, Japan) to count TRAP-positive multinucleated cells with three or more nuclei, and the number was graphed. Data were derived from three independent experiments.

### 4.4. Pit Formation Assay

RAW 264.7 cells (5 × 10^4^ cells/well) were seeded in a Corning^®^ osteo assay surface (Corning, New York, NY, USA), which was coated with a synthetic surface made of inorganic crystal-line calcium phosphate, cultured in a differentiation medium containing 10 ng/mL of RANKL (DMEM including 10% FBS), and treated with various concentrations (100 to 500 μM) of melatonin. The medium was replaced daily for 5 days. After 5 days of culture, the osteoclasts formed in each well were removed using a 5% sodium hypochlorite solution and washed twice with distilled water. Resorption areas of the osteo assay surface by osteoclasts were captured using a digital camera (Nikon Coolpix, Tokyo, Japan) attached to an Olympus CKX41 inverted optical microscope (Olympus, Tokyo, Japan). The total area and absorption area of each image were quantified and graphed using Adobe Photoshop CS6 software (Adobe Systems, San Jose, CA, USA).

### 4.5. Western Blot Analysis

RAW 264.7 cells (1 × 10^6^ cells) were seeded in 100 mm culture dishes and incubated for 2 h in a CO_2_ incubator to allow the cells to adhere to the bottom of the culture dishes. After 2 h, the medium was replaced with a differentiation medium containing 10 ng/mL of RANKL (DMEM including 10% FBS). Cells were treated with 300 μM melatonin and incubated for a minimum of 5 min and a maximum of 5 days. At the end of each treatment, cells were harvested and lysed using RIPA lysis buffer (Sigma-Aldrich, Louis, MO, USA). The protein concentration was determined using a protein assay kit (Bio-Rad, Budapest, Hungary) according to Bradford’s method for protein quantification [48]. The amount of protein was 20 µg and each sample was loaded on a 10% SDS-PAGE gel and transferred to a polyvinylidene difluoride (PVDF) membrane (Amersham GE Healthcare, Little Chalfont, UK). Then, the membrane was blocked using 5% non-fat dried milk. The primary antibody (1:1000) was reacted at room temperature for 1 h; the antibodies used in this study are as follows: ERK (Cell Signaling Technology, Beverly, MA, USA), phospho-ERK (Cell Signaling Technology, Beverly, MA, USA), JNK (Cell Signaling Technology, Beverly, MA, USA), phospho-JNK (Cell Signaling Technology, Beverly, MA, USA), NFATc1 (Cell Signaling Technology, Beverly, MA, USA), TRAF6 (Cell Signaling Technology, Beverly, MA, USA), c-Fos (Cell Signaling Technology, Beverly, MA, USA), DC-STAMP (Abcam, Cambridge, MA, USA), MMP9 (Abcam, Cambridge, MA, USA), cathepsin K (Abcam, Cambridge, MA, USA), ATP6v0d2 (Abcam, Cambridge, MA, USA), and β-actin (Santa Cruz Biotechnology Inc., Santa Cruz, CA, USA). Thereafter, the membrane was washed with PBS-T (phosphate-buffered saline with Tween 20) 5 times for 10 min each. HRP-conjugated secondary antibody (1:5000) was reacted at room temperature for 1 h and then washed with PBST 5 times for 10 min each (mouse anti-rabbit IgG (Enzo Life Sciences, Farmingdale, NY, USA) and rabbit anti-mouse IgG (Enzo Life Sciences, Farmingdale, NY, USA)). SuperSignal™ West Femto Maximum Sensitivity Substrate (Thermo Scientific, Rockford, IL, USA) was used to perform immunoblotting and Image Quant LAS 500 (GE Healthcare, Chicago, IL, USA) was used to detect the protein expression.

### 4.6. Immunofluorescence Analysis

RAW 264.7 cells were seeded on a Nunc™ Lab-Tek™ II Chamber Slide™ System (Nunc; Thermo Fisher Scientific, Rochester, NY, USA) and incubated for 2 h in a CO_2_ incubator to allow the cells to adhere to the bottom of the slide well. Then, cells were applied to a differentiation medium and 300 µM of melatonin. The medium was replaced daily for 5 days. Cells were fixed in 4% formaldehyde solution (Sigma-Aldrich, Louis, MO, USA) for 10 min, and 0.2% Triton X-100 was applied for 10 min at room temperature to allow the antibody to penetrate the cells. Subsequently, to block nonspecific binding, the cells were blocked with PBS including 1% bovine serum albumin (BSA) for 1 h. The cells were incubated in the primary antibody (diluted at 1:100; NFATc1 and TRAF6 (Cell Signaling Technology, Beverly, MA, USA)) at 4 °C overnight. The cells were washed 3 times with PBS, then the secondary antibody (diluted at 1:200; Alex Fluor 488 anti-mouse; Invitrogen Gaithersburg, MD, USA) was incubated for 1 h at room temperature. Then, rhodamine phalloidin (Invitrogen, Eugene, OR, USA) was applied for 30 min to stain actin, and DAPI (Invitrogen, Eugene, OR, USA) was applied for 10 min to stain the nucleus. The expression of each protein was scanned and analyzed using a Zeiss LSM 750 laser scanning confocal microscope (Göttingen, Germany).

### 4.7. ATPase Activity

RAW264.7 cells (5 × 10^4^ cells) were seeded on a 24-well tissue culture plate for 2 h in a CO_2_ incubator. Then, cells were applied to a differentiation medium and treated with (minimum of 100 and a maximum of 500 µM) melatonin for 1, 3, and 5 days. After treatment, cells were washed twice with PBS and applied with an ATPase/GTPase Activity Assay Kit (Sigma-Aldrich, Louis, MO, USA) according to the manufacturer’s instructions. The supernatant was transferred to a 96-well plate, and the absorbance of each well was measured at a wavelength of 620 nm using a SpectraMax iD3 microplate reader (Molecular Devices, Sunnyvale, CA, USA). Data were derived from at least 3 independent experiments.

### 4.8. Real-Time PCR

RAW264.7 cells (2 × 10^5^ cells) were seeded on a 60-mm culture dish for 2 h in a CO_2_ incubator. Then, cells were applied to a differentiation medium and treated with 300 µM of melatonin for 1, 3, and 5 days. Total RNA and cDNA were prepared using the RNeasy Mini Kit (Qiagen, Hilden, Germany). The total RNA concentration was determined with a NanoDrop 2.0 spectrophotometer (Thermo Fisher Scientific, Pittsburgh, PA, USA). A quantitative real-time PCR was performed with Maxima™ SYBR Green/ROX qPCR Master Mix (Fermentas) and run on an ABI 7500 Fast Real-Time PCR System (Applied Biosystems 7500 System, Foster City, CA, USA) using Sequence Detection System software version 2.0.1. The relative mRNA levels were normalized using GAPDH as a housekeeping gene. Atp6v0d2 (NM_152565) and GAPDH (NM_008084) messenger RNA expressions were quantified through QuantiTect Primer Assays (Qiagen, Hilden, Germany).

### 4.9. Small Interfering RNA (siRNA) Transfection

RAW264.7 cells (2 × 10^5^ cells) were seeded in 6-well plates in medium for 2 h in an incubator. NFATc1-siRNA or uncorrelated siRNA (si-control) were purchased from Qiagen (Germantown, Maryland, USA) and used. Transfection of siRNA was performed using Lipofectamine RNAiMAX (Invitrogen, Thermo Fisher Scientific Carlsbad, CA, USA) and SiRNA transfected RAW264.7 cells in Opti-MEM (Thermo Fisher Scientific Carlsbad, CA, USA) medium (without FBS and antibiotics) according to the manufacturer’s protocol. After 6 h, RANKL (10 ng/mL) and melatonin 300 were added to the medium. One day after transfection, cells were harvested and protein expression was confirmed by western blot.

### 4.10. Statistical Analysis

All data were derived from three independent experiments and expressed as mean ± standard deviation. To statistical analysis, GraphPad Prism software version 5.0 (San Diego, CA, USA) was used and one-way ANOVA and a paired sample t-test were used to analyze data on cell viability, TRAP positivity, and pit formation as well as real-time PCR.

## Figures and Tables

**Figure 1 molecules-27-00501-f001:**
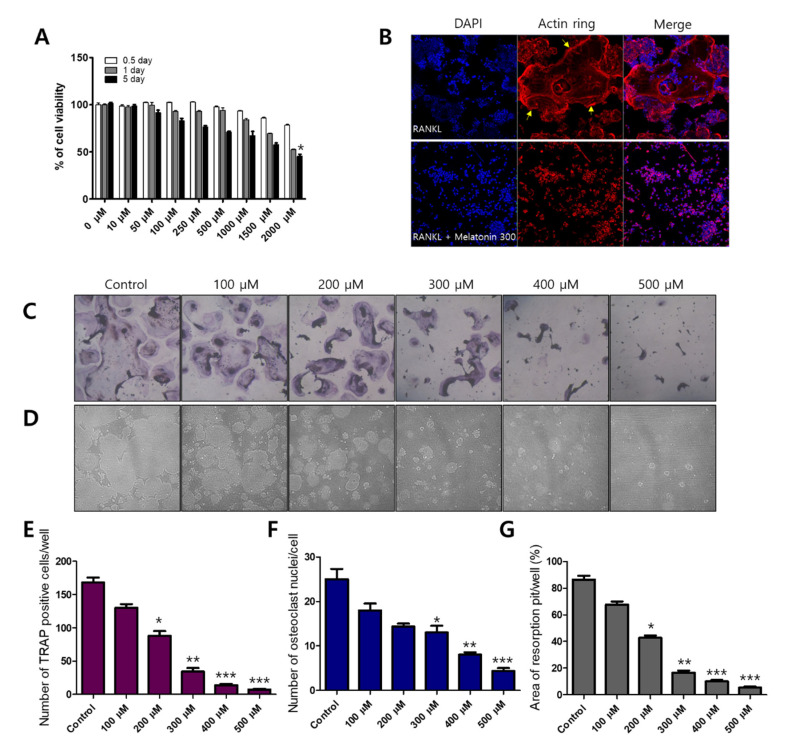
Inhibitory effect of melatonin on RANKL-induced osteoclastogenesis. (**A**) Cell viability of RAW 264.7 cell macrophages treated with various concentrations of melatonin for 0.5 to 5 days according to MTT assay. (**B**) RAW264.7 cells were cultured with RANKL or RANKL + melatonin for 5 days. Cells were stained with fluorescein rhodamine phalloidin to detect the presence of actin rings. The yellow arrow indicates the edge of the actin ring. (**C**,**E**) Number of TRAP-positive multinucleated RAW 264.7 cells containing three or more nuclei after treatment with 10 ng/mL RANKL and 100 to 500 μM melatonin. (**F**) Number of osteoclast nuclei. (**D**,**G**) Resorption pit formation and resorption areas expressed as area of pit formation with respect to total area. Values represent the means of three independent experiments ± SDs (*n* = 6). * *p* < 0.05, ** *p* < 0.01, and *** *p* < 0.001 compared with the control.

**Figure 2 molecules-27-00501-f002:**
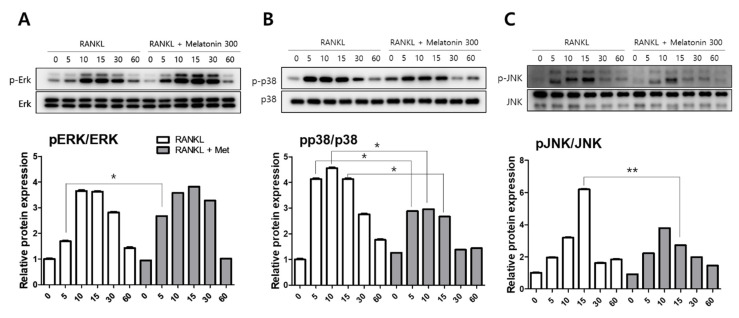
Effects of RANKL and melatonin treatment on MAPK signaling in RAW264.7 cells. Cytosolic protein levels of (**A**) p-ERK, (**B**) p-p38, and (**C**) p-JNK in RAW 264.7 cells treated with RANKL (10 ng/mL) and RANKL (10 ng/mL) + melatonin (300 μM) for 5, 10, 15, 30, and 60 min according to western blot analysis. The expression data were normalized to the total MAPK (ERK, p38, and JNK) signal. Values represent the means of three independent experiments ± SDs. RANKL alone treatment and RANKL + melatonin combination treatment were verified using a paired sample *t*-test (* *p* < 0.05, ** *p* < 0.01).

**Figure 3 molecules-27-00501-f003:**
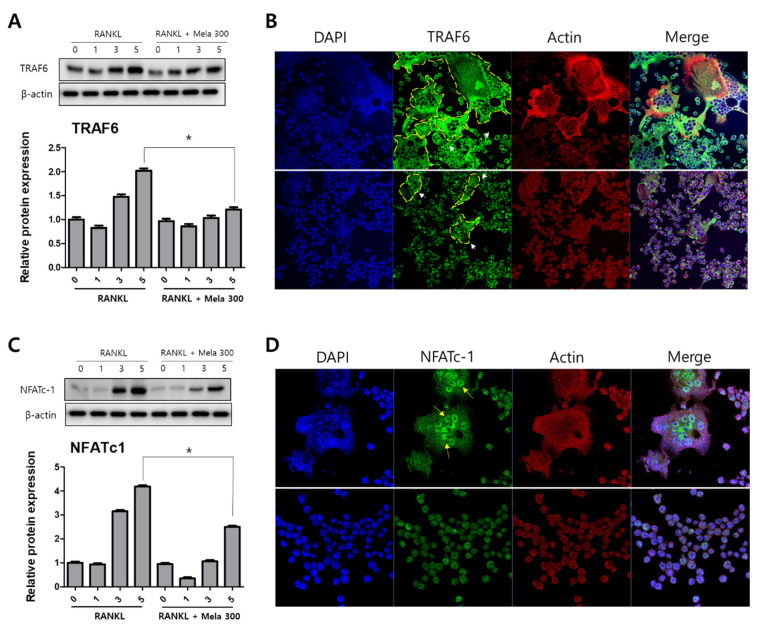
Reduction of TRAF6 and NFATc1 protein expression in RAW264.7 cells following RANKL and RANKL + melatonin treatment for 1, 3, and 5 days. (**A**,**C**) Protein expression of TRAF6 and NFATc1 according to western blot analysis. The expression data were normalized using β-actin. Values represent the means of three independent experiments ± SDs. RANKL alone treatment and RANKL + melatonin combination treatments were verified using a paired sample *t*-test (* *p* < 0.05). (**B**,**D**) Confocal immunofluorescence images of TRAF6 and NFATc1 (green) staining and nuclear (blue) and actin (red) counterstaining of RAW264.7 cells incubated for 5 days with RANKL and RANKL + melatonin, respectively.

**Figure 4 molecules-27-00501-f004:**
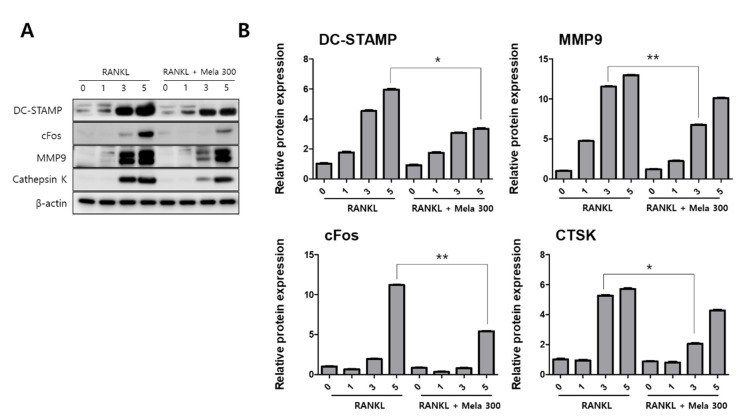
Suppression of the terminal stage and cell-cell fusion markers in osteoclast differentiation in RAW264.7 cells following RANKL and RANKL + melatonin treatment for 1, 3, and 5 days. (**A**) Expression of osteoclastogenesis-related proteins according to western blotting. (**B**) Protein expression graphs of proteins displayed in 4A. The expression data were normalized using β-actin. Values represent the means of three independent experiments ± SDs. RANKL alone treatment and RANKL + melatonin combination treatment were verified using a paired sample t-test (* *p* < 0.05, ** *p* <0.01).

**Figure 5 molecules-27-00501-f005:**
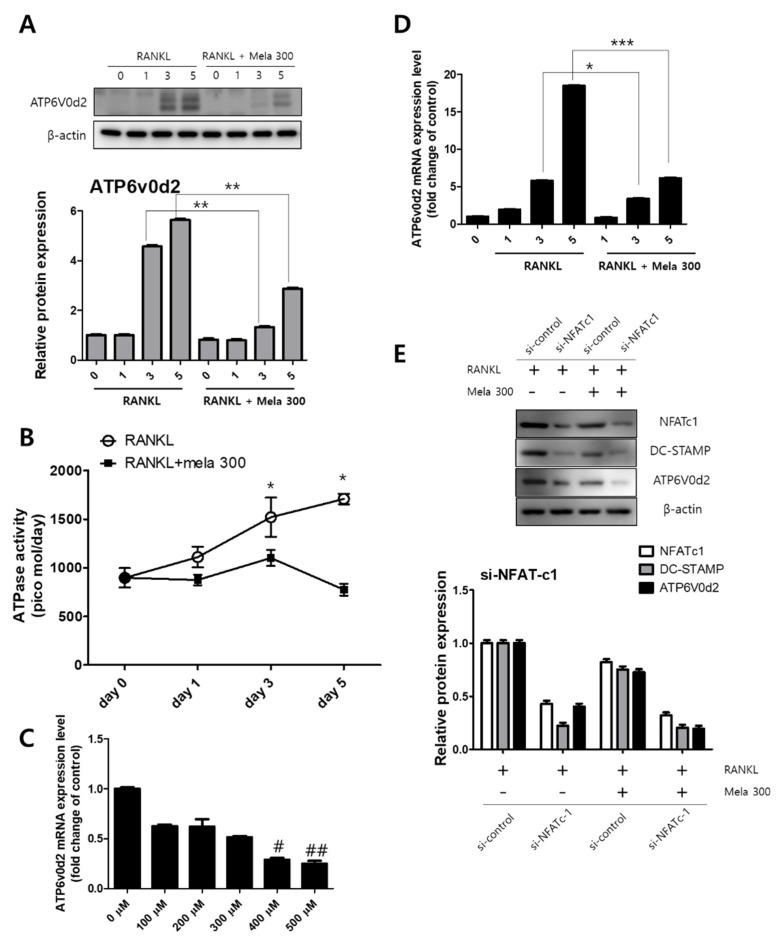
Regulation of V-ATPase (Atp6v0d2) expression following (**A**,**D**) treatment of RAW264.7 cells with RANKL and RANKL + melatonin for 1, 3, and 5 days. (**A**) Graph of Atp6v0d2 expression according to western blotting. (**D**) mRNA expression of Atp6v0d2 according to real-time PCR. Values represent the means of three independent experiments ± SDs RANKL alone treatment and RANKL + melatonin combination treatments were verified using a paired sample t-test (* *p* < 0.05, ** *p* < 0.01, and *** *p* < 0.001). (**B**) ATPase activity was conducted in RAW264.7 cells treated with RANKL (10 ng/mL) and melatonin (100 to 500 μM) for 1 to 5 days and (**C**) Atp6v0d2 gene expression of RAW264.7 cells treated with RANKL (10 ng/mL) + melatonin (100 to 500 μM) for 24 h. (**E**) After transfection of si-NFATc1 into RAW264.7 cells, RANKL and RANKL + melatonin 300 were treated for 1 day. NFATc1 knockdown using small molecule interfering (si) RNA decreased the total amount of DC-STAMP and ATP6V0d2. Values represent the means of three independent experiments ± SDs (# *p* < 0.05 and ## *p* < 0.01 compared with the control).

## Data Availability

Data will be available from the corresponding author upon logical request.

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
