# Peer review of "Melatonin Attenuates RANKL-Induced Osteoclastogenesis via Inhibition of Atp6v0d2 and DC-STAMP through MAPK and NFATc1 Signaling Pathways"

_molecules, 2022, doi:10.3390/molecules27020501_

Round 1

Reviewer 1 Report

To the best of my knowledge, the findings presented here are largely confirmatory and essentially validate prior published studies, including 33713122, 28587149, 33452455… (PMID). Although decreased the function of osteoclastogenesis via melatonin is interesting, in light of findings from other literatures, the affected MAPK and NFATc1 signaling pathways is rather expected. Moreover, the clinical trials had been conducted postmenopausal women with osteopenia using nightly placebo or melatonin (NCT01152580, NCT01690000, NCT01870115). Therefore, the novelty of this manuscript is low. More novel scientific findings are needed! 

Author Response

Dear Editor and reviewers of 'Molecules'

First of all, we would like to express our gratitude to everyone who recognized the value of this manuscript and contributed to the review. We would also like to thank the editors for giving us another chance to submit this manuscript. It would be a great honor for us if this manuscript could be published in 'Molecules'.
We are looking forward to having your thoughtful decision.
Sincerely yours,
Corresponding Author :  In-Ryoung Kim  

Reviewer 2 Report

This manuscript by Kim et al., explores the effects of melatonin on RANKL-induced osteoclastogenesis and functional properties of RAW 264.7 cells. The authors report that exogenous melatonin reduces osteoclast differentiation and expression of osteoclast-related genes as well as bone resorption activity. In addition, it is concluded that melatonin inhibits osteoclast differentiation and cell fusion by inhibiting the expression of Atp6v0d2 through the inactivation of MAPK and NFATc1 signaling pathways.

The topic is interesting and the authors managed to draw their conclusions using a variety of experimental approaches. However, the following points need to addressed:

  1. The introduction should be more focused on the role of melatonin in bone metabolism. The concepts of the balanced skeletal remodelling and the regulation of osteoclastogenesis are well known; a more concise description of these with a more extended description of melatonin actions on bone cells would add to a more informative introduction.
  2. Although the inhibitory effect of melatonin on RAW 264.7 cell fusion to form osteoclasts and subsequent pit formation is reported here, to assess the osteoclastic activity, an actin-ring formation analysis is essential. This will provide useful information for the effects of melatonin on osteoclast proper or not activity (integral or fragmented actin ring formation, respectively). The fact that the osteoassay used contains only CaP crystalline salts and not properly form hydroxyapatite with collagen type I, contributes to the necessity of performing this analysis.
  3. Results 2.2. Since the differences were detected in different time points, e.g. 5min for pERK and 15min for pJNK, should be mentioned in the results and also discussed.
  4. Results 2.2 (cont.). The finding that MMP-9 and cathepsin K levels are still rising in melatonin-treated cells after 5d in culture seems that contradicts with the previous results, since these are the main proteolytic enzymes that degrade bone matrix. Please explain and comment. Are these enzymes active? For example, a zymogram would provide an answer to this for MMP-9.
  5. The ATPase activity must be expressed as a function of time. Fig5A, time points are not aligned with lanes in the WB.
  6. Text needs some editing, e.g. lines 158-163.

Author Response

(The authors gave the same response as above.)

Round 2

Reviewer 1 Report

The authors have accordingly revised the manuscript.

Author Response

Thank you for your heartfelt advice and help to increase the value of this manuscript.

Reviewer 2 Report

All points were addressed sufficiently. The only point that needs to be corrected/change is the ATPase activity units. This should be expressed as pM/min or pM/sec or anything similar (quantity over time) and not per well, since the enzyme activity is measured.

Author Response

Thanks for the sincere advice. Following your advice, the units used in Figure 5B have been corrected to pM/day. Thank you for helping to increase the value of this manuscript.